# Application of Whole Exome Sequencing and Functional Annotations to Identify Genetic Variants Associated with Marfan Syndrome

**DOI:** 10.3390/jpm12020198

**Published:** 2022-02-01

**Authors:** Min-Rou Lin, Che-Mai Chang, Jafit Ting, Jan-Gowth Chang, Wan-Hsuan Chou, Kuei-Jung Huang, Gloria Cheng, Hsiao-Huang Chang, Wei-Chiao Chang

**Affiliations:** 1Department of Clinical Pharmacy, School of Pharmacy, Taipei Medical University, Taipei 110, Taiwan; jennlin@tmu.edu.tw (M.-R.L.); jkt5265@tmu.edu.tw (J.T.); ocean.chou@tmu.edu.tw (W.-H.C.); m301105029@tmu.edu.tw (K.-J.H.); 2Ph.D. Program in Medical Biotechnology, College of Medical Science and Technology, Taipei Medical University, Taipei 110, Taiwan; awefld@gmail.com; 3Center for Precision Medicine, China Medical University Hospital, Taichung 404, Taiwan; d6781@mail.cmuh.org.tw; 4School of Medicine, China Medical University, Taichung 404, Taiwan; 5USC Dornsife College of Letters, Arts and Sciences, University of Southern California, Los Angeles, CA 90007, USA; glory132@gmail.com; 6Department of Surgery, School of Medicine, Taipei Medical University, Taipei 110, Taiwan; 7Division of Cardiovascular Surgery, Department of Surgery, Taipei Veterans General Hospital, Taipei 112, Taiwan; 8Master Program for Clinical Pharmacogenomics and Pharmacoproteomics, School of Pharmacy, Taipei Medical University, Taipei 110, Taiwan; 9Integrative Research Center in Critical Care, Wan Fang Hospital, Taipei Medical University, Taipei 116, Taiwan

**Keywords:** Marfan syndrome, whole-exome sequencing, new mutations, *FBN1*, *TTN*, *POMT1*

## Abstract

Marfan syndrome (MFS) is a rare disease that affects connective tissue, which causes abnormalities in several organ systems including the heart, eyes, bones, and joints. The autosomal dominant disorder was found to be strongly associated with *FBN1*, *TGFBR1*, and *TGFBR2* mutations. Although multiple genetic mutations have been reported, data from Asian populations are still limited. As a result, we utilized the whole exome sequencing (WES) technique to identify potential pathogenic variants of MFS in a Taiwan cohort. In addition, a variety of annotation databases were applied to identify the biological functions as well as the potential mechanisms of candidate genes. In this study, we confirmed the pathogenicity of *FBN1* to MFS. Our results indicated that *TTN* and *POMT1* may be likely related to MFS phenotypes. Furthermore, we found nine unique variants highly shared in a MFS family cohort, of which eight are novel variants worthy of further investigation.

## 1. Introduction

Marfan syndrome (MFS) is a connective tissue disease with an estimated prevalence rate of 10.2/100,000 in Taiwan [1]. Typical characteristics of MFS patients include a tall and thin stature, long limbs, pectus, lens dislocation, overly flexible joints, and scoliosis. Despite ocular and musculoskeletal deformities, the disease most significantly manifests in the cardiovascular system. Valvular disorders and aortic aneurysm are common among patients, which may result in the progression of aortic dissection, and the rupture of the aortic root has been the leading cause of mortality [2]. Since there is currently no cure for MFS, treatment strategies focus mostly on preventing fatal complications. Prophylactic medications such as β-blockade, calcium channel blockers (CCB), or angiotensin-converting enzyme inhibitors (ACEI) have been used to reduce hemodynamic stress on the aortic root. Surgical replacement of the heart valves or aortic root is necessary for patients with a more severe condition [3]. Diagnosis of MFS relies mainly on Ghent nosology. It provides a set of defined criteria to describe the clinical signs, family history, and pathogenic mutations [4]. Transthoracic echocardiography [5] and magnetic resonance imaging [6] could be ordered in addition to assess cardiovascular abnormalities. With the advancement of molecular diagnostics, physicians can also identify potential MFS patients more efficiently and accurately by searching known mutation sites [7].

Previous studies have found *FBN1*, *TGFBR1*, and *TGFBR2* to be notable genes associated with the pathogenesis of MFS. *FBN1* encoding fibrillin-1 is an essential protein for forming microfibrils. Mutations in *FBN1* results in the malformation of the extracellular matrix, which leads to the decrease in integrity and function of connective tissues. MFS patients showed highly variable mutation types in the *FBN1* gene, but cysteine substitution has known to be the most common class of mutation among MFS patients [8,9]. More recently, *TGFBR1* and *TGFBR2* were found to be linked to progressive dilation of the aortic root [10,11]. *TGFB2* heterozygous knockout mice presented a dilation compared to normal mice by eight months, demonstrating that a single allele is sufficient to be pathogenic [12].

Even though patterns of genetic inheritance have been reported for MFS, only a few were from Asian populations. Therefore, in the current study, we aimed to utilize whole exome sequencing (WES) technique to investigate the genetic factors associated with MFS. We particularly focused on the potential pathogenic rare variants shared in a family cohort to investigate the hidden unique inheritance pattern within a Taiwanese MFS family. In addition, functional annotation databases including knockout mouse phenotypes, Gene Ontology (GO), and KEGG pathways were used to gain further insights into the biological level and possible disease mechanisms. Our study workflow is illustrated in Figure 1.

## 2. Results

### 2.1. Patient Basal Characteristic

We recruited 10 MFS patients and one healthy volunteer including five females and six males (Table 1). Participants were between 20 to 52 years old, with a mean age of 37.7 years old. Out of the 11 subjects, five individuals were a family cohort with one family member acting as the healthy control.

### 2.2. Pathogenic Rare Variants on Previously Reported MFS-Related Genes

First, we collected the discovered genes related to MFS from previous studies (Appendix A). We hypothesized that some of the established results could be replicated through WES analysis in our MFS cohort. As shown in Table 2, a total of 19 rare variants were identified on eleven previously reported MFS-related genes (Table 2, Appendix A). The 19 rare variants were further filtered through the ClinVar and CADD/REVEL score for pathogenicity predictions. Four pathogenic rare variants, chr9:134758250 on *COL5A1*, chr15:48448860 and chr15:48503803 on *FBN1*, and chr16:15714999 on *MYH11*, were identified by ClinVar; three additional variants, chr15:48436991 and chr15:48448860 on *FBN1*, and chr19:8136499 on *FBN3*, were predicted to be pathogenic by CADD/REVEL score. We found that chr15:48448860 on *FBN1* was predicted to be pathogenic by both criteria. Three of the six identified pathogenic rare variants were located on *FBN1*. We further observed from the clinical data that seven (pt.3, 4, 6, 7, 8, 9, 11) out of ten MFS patients carried one *FBN1* variants (Table 2); pt.1 who carried two variants on *FBN1* had the highest systematic score (Appendix A). Furthermore, none of the *FBN1* variants were found in the healthy subject (pt.10). These results aligned with previous findings that the penetrance for *FBN1* is typically high, and carrying more *FBN1* variants tends to correlate with a more severe clinical manifestation (Appendix A).

### 2.3. Functional Annotation of MFS Candidate Genes

Through screening the sequencing data with the selection criteria described in the method, we included 11 MFS-related genes, 66 genes were generated by ClinVar pathogenicity results, and 151 genes from CADD and REVEL annotations. After excluding the overlapping genes in three categories, a total of 219 genes were defined as “MFS candidate genes” and proceeded to enrichment analysis (Appendix A). The significance threshold was set at *q*-value (FDR) < 0.05 for each functional annotation.

We first performed an over-representation analysis (ORA) based on the mammalian phenotype ontology database to determine the phenotypic impact of the candidate genes in mouse models. A total of 219 human gene names were converted to 229 mouse gene names for further analysis. Eight knockout mouse phenotypes were significantly enriched (Table 3). The top significant phenotype was “abnormal cardiovascular system morphology”, which is frequently observed in MFS patients. We also found several other cardiovascular phenotypes and fatal disorders enriched in MFS candidate genes. The Kyoto Encyclopedia of Genes and Genomes (KEGG) was used to perform molecular pathway enrichment analysis. However, no pathway was significantly enriched in this category. Furthermore, Gene Ontology (GO) annotations were used to evaluate protein–protein interaction. A total of 15 GO terms were significantly enriched (Table 3). Among all, “sensory perception of light stimulus” was the most significant term. Patients with MFS usually suffer from nearsightedness, with a thinner and flatter cornea [13]. A cohort study also reported a correlation between MFS and the thinning of the retinal nerve fiber layer (RNFL) [14]. Progressive loss of RNFL would lead to a decrease in retinal ganglion cells and cause visual function impairment [15]. Moreover, we observed three GO terms, “extracellular structure organization”, “extracellular matrix structural constituent”, and “extracellular matrix”, which are all related to ECM in three different aspects. MFS is a connective tissue disorder caused by mutations in the ECM protein, indicating ECM organization is crucial.

### 2.4. Pathogenic Rare Variants on Candidate MFS Genes

A total of 74 rare variants on 66 genes, and 167 rare variants on 151 genes were determined as pathogenic, according to the ClinVar and CADD/REVEL score, respectively. Among all pathogenic rare variants predicted by ClinVar, four were located on *TTN* and three were on *PRSS1* (Figure 2a). It is noteworthy that the *TTN* gene encodes titin, which has been reported to associate with muscular [16] and cardiomyopathy diseases [17], implying a potential relationship between *TTN* and MFS. Most of the genes predicted by CADD/REVEL score included only one pathogenic variant, while *ATP11B* and *SHPRH* each had three variants (Figure 2b). Importantly, three variants were found pathogenic by both the ClinVar database and CADD/REVEL scores including chr9:131513282 on *POMT1*, chr15:48448860 on *FBN1*, and chr17:75831135 on *UNC13D* (Table 4). *POMT1* was reported to be important in a series of metabolic diseases called muscular dystrophy-dystroglycanopathies (MDDGs), accompanied by variable degrees of intellectual disability, with brain and ocular abnormalities [18]. This *FBN1* variant was reported to associate with MFS [19] and familial thoracic aortic aneurysm and aortic dissection (TAAD) [20]. Mutations on *UNC13D* lead to defects in the cell destruction process, which cause a severe inflammatory syndrome called hemophagocytic lymphohistiocytosis [21].

### 2.5. The Inheritance Pattern within a Taiwanese MFS Family Cohort

A family cohort of five subjects was recruited in the study, consisting of father (pt.10), mother (pt.7), two sons (pt.8 and 9), and a daughter (pt.11). The father is a healthy subject while the other four individuals were MFS patients. As shown in Table 5, nine nonsynonymous single nucleotide variations of nine genes predicted by ClinVar or CADD/REVEL score were highly shared (≥75%) in the patients. Two variants on *ALAS1* and *FBN1* were shared in all four MFS patients.

### 2.6. Pathogenic Rare Variants Carried by Patient no.2 and no.5

Most MFS patients in this study carried at least one *FBN1* variant, except for pt.2 and pt.5. Thus, we examined their variants to investigate whether any novel pathogenic variations are from the two patients. Variants carried by the healthy subject were excluded from analysis. Through ClinVar and CADD/REVEL score prediction, we identified 23 variants of 22 genes in pt.2, none of which were found to be associated with MFS in the previous studies (Appendix A). By querying the GTEx portal, we found that *RBM20* and *PKP2* were highly expressed in the heart. *LPIN1* was expressed in skeletal muscle. *TTN*, the pathogenic variant predicted in the previous section, was also found to be highly expressed in both the heart and muscle (Figure 3a). On the other hand, a total of 21 pathogenic variants on 21 genes were carried by pt.5 (Appendix A). Among all genes, CAVIN4 (MURC) was highly expressed in skeletal muscle and heart tissue (Figure 3b). Surprisingly, hardly any genes carried by pt.2 and pt.5 were overlapped, indicating that the genetic pathology of these two patients were distinct. Further investigation is required to confirm the genetic pathology in patients 2 and 5.

## 3. Discussion

Our study investigated the pathogenic variants within 10 MFS patients and one healthy subject through WES and identified the biological functions of MFS candidate genes. Most pathogenic variants on MFS-related genes were located on *FBN1.* Eight of the ten patients were found to carry at least one *FBN1* variant. Thus, the critical role of *FBN1* for MFS was confirmed in the Taiwanese population. We further found nine variants highly shared in a MFS family, of which eight were on novel genes and one was on *FBN1* with its pathogenic variant not previously reported. Furthermore, the two pathogenic variants on *FBN1* and *ALAS1* were shared in all four patients. *ALAS1* encodes the mitochondrial enzyme that catalyzes the rate-limiting step in heme biosynthesis and is associated with acute porphyria and sideroblastic anemia [22]. A recent study showed that *ALAS1* heterozygous mice had an age-dependent reduction of free heme in skeletal muscle [23]. In addition, a case report study also indicated that MFS patients might present anemia after aortic dissection surgery [24]. However, the direct association between MFS and *ALAS1* variants need further investigation.

Among all the MFS candidate genes we identified, *TTN* and *POMT1* are considered as the most crucial genes (Figure 2a, Table 4 and Table 5). *TTN* mainly expresses in the heart and the muscle according to the GTEx portal. It is responsible for encoding sarcomere, the largest muscle filament in the heart. Consistent with previous studies, genetic variants on *TTN* have been reported to be associated with heart conditions [25]. Moreover, four pathogenic variants were found in *TTN*, while most genes we identified included only one pathogenic variant. The *TTN* variant was also highly shared in the family cohort. Likewise, *POMT1* is considered as an important target because one of its variants, rs146869947, was observed in 75% of the MFS family members. This variant was also predicted to be pathogenic by both ClinVar and CADD/REVEL scoring tools. Although the underlying mechanism remains unclear, *POMT1* was reported to associate with the presentation of cardiomyopathy among patients with limb-girdle muscular dystrophy [26].

With regard to patients whose genetic defects cannot be explained by *FBN1*, we utilized the GTEx portal to investigate genes that are particularly expressed in heart or muscle tissues. For patient 2, we focused on *PKP2*, *LPIN1*, *RBM20*, and *TTN*. *PKP2* encodes plakophilin-2, which is a component of the desmosome and its pathogenic role has been recognized in inherited cardiac arrhythmias syndromes [27]. *LPIN1* plays a role in lipid synthesis and storage, and mutation in *LPIN1* is associated with myoglobinuria as well as rhabdomyolysis [28]. *RBM20* and *TTN* are a pair of genes that are strongly related to dilated cardiomyopathy. *RBM20* regulates the splicing event of *TTN.* The mis-spliced exon is located in the elastic PEVK region in the I-band of the heart, and thus leads to an increased elasticity of the sarcomere [29]. For patient 5, *CAVIN4* (*MURC*) was considered as a potential candidate gene since it reaches a certain expression level in the heart and muscle. *CAVIN4* (*MURC*) modulates cardiac muscle cell signaling and myofibrillar organization [30]. It was reported to be associated with pulmonary hypertension [31] and familial dilated cardiomyopathy [32]. Therefore, the results implied that *PKP2*, *LPIN1*, *RBM20*, *TTN*, and *CAVIN4* (*MURC*) may be related to the MFS phenotypes of patient 2 and 5, respectively.

While our study illustrates the power of applying WES to identify disease causing genes in a rare disease cohort, it also displayed several limitations. First, we found that patient 1, who carried two *FBN1* variants, had the highest systemic score. However, some patients with no *FBN1* variant still presented higher scores compared to other patients with the variant. As a result, a larger sample size is needed in order to construct a MFS genotype–phenotype correlation map. Second, we have only recruited one family cohort thus far. More samples from family cohorts will be helpful to verify the variants and to construct pedigrees in order to identify the disease-causing genes precisely. Third, rare variants we included must present a MAF below 0.01 of the East Asian population in multiple databases. Although this is a relatively conservative strategy, we might miss some variants that is particularly rare in the Taiwanese population, but not in other East Asian populations. Therefore, this approach might ignore the variants that are likely to be unique and important for Taiwanese MFS patients.

To understand the biological functions of our candidate genes, we performed three functional annotations. The results strongly resemble the clinical manifestations of MFS, which strengthens the confidence in our candidate genes and further support the feasibility of utilizing the WES analysis for variant identification in rare diseases. In conclusion, by the use of whole exome sequencing, we confirmed the pathogenicity of the *FBN1* gene to MFS in the Taiwanese patients. Our results also implied that *TTN* and *POMT1* are likely to be the most potential candidate genes, supported by their important biological functions in heart and muscle. Furthermore, we found nine unique variants highly shared in a MFS family cohort, of which eight were novel variants that are worth further investigation.

## 4. Materials and Methods

### 4.1. Study Subjects and Sample Preparation

All subjects were recruited from Taipei Veterans General Hospital. The Revised Ghent criteria was used for the diagnosis of Marfan syndrome [4]. All of the index cases and their family who were diagnosed as Marfan syndrome were made by two doctors. One was in charge of the index case, the other a pediatric geneticist. Informed consent was obtained from all participants before the initiation of the study. This study was approved by the Institutional Review Board (IRB) of Taipei Veterans General Hospital (2018-01-005B). Peripheral whole blood samples were collected from all subjects at Taipei Veterans General Hospital and transferred to Taipei Medical University for genomic DNA isolation. Nine of the DNA samples were extracted by the phenol-chloroform method and the other two were isolated using QIAamp DNA Blood Maxi Kit (QIAGEN, Germantown, MD, USA).

### 4.2. Whole Exome Sequencing and Bioinformatic Analysis

We analyzed samples from 11 patients by PCR and Nextera whole exome sequencing (WES) using genomic DNA as a template. Paired-end sequencing was performed on NovaSeq 6000 platform, with sequencing depth above 50× (6G). Sequence capture, enrichment, and elution were performed according to the manufacturer’s instruction and protocols. All raw sequences were analyzed through the GATK germline short variant discovery pipeline. Adapters and low quality reads were trimmed using Trimmomatic under the PE module (ILLUMINACLIP: TruSeq3-PE-2.fa: 2:30:10 LEADING:3 TRAILING:3 SLIDINGWINDOW:4:15 MINLEN:36). Clean sequences were then aligned to the human reference genome GRCh38 using Burrows–Wheeler Aligner with default parameters. GATK was used to remove PCR duplicates and perform base quality score recalibration. Nonsynonymous single nucleotide variants, insertions, and deletions were called by GATK HaplotypeCaller. Rare variants with minor allele frequency <0.01 in Taiwan Biobank [33], 1000 Genomes Project [34] (1KGP; East Asian), The Genome Aggregation Database [35] (gnomAD; East Asian), and The Exome Aggregation Consortium [36] (ExAC; East Asian) were selected. Finally, functional annotations were performed with ANNOVAR to interpret the information of these variants. Pathogenicity was determined using clinical significance annotations on: (A) ClinVar (v.20210501) with variants annotated with either (1) association, (2) conflicting interpretations of pathogenicity, (3) likely pathogenic, (4) pathogenic, or (5) pathogenic/likely pathogenic; (B) variants with CADD Phred scores >20, indicating the variant is in 1% of the most pathogenic variants, and REVEL raw scores ≥0.75, predicted as pathogenic variants with high specificity but low sensitivity. These criteria were used to filter the pathogenic variants from the pool of rare variants.

### 4.3. Marfan Syndrome Related Genes

We recruited 18 studies [37,38,39,40,41,42,43,44,45,46,47,48,49,50,51,52,53,54] related to MFS, and defined 26 genes as “MFS-related genes” (Appendix A). Studies were accessed from PubMed on 27 March 2020, according to the keywords “Marfan”, “gene”, “variants”, “mutations”, “association”, “Marfan syndrome”, “Marfan-like disease”, “Loeys–Dietz syndrome”, “Vascular Ehlers–Danlos syndrome”, and “Thoracic Aortic Aneurysm and Dissection (TAAD)”. Genes reported to be associated with the onset of MFS, MFS severity, MFS susceptibility, or related to MFS-like diseases were enrolled in our study. Gene List Automatically Derived For You (GLAD4U) was used to help prioritize the gene list [55].

### 4.4. Functional Annotation

A list of candidate genes was selected after WES analysis. The inclusion criteria were depicted as: (1) genes with at least one potential pathogenic rare variant and (2) MFS-related genes identified in our sequencing data. Three gene-based annotations were conducted by the WebGestalt 2019 functional enrichment analysis web tool [56] to gain mechanistic insights into our gene list. (1) Knockout mouse phenotype: genes were first converted to the mouse gene name using BioMart [57]. The Mammalian Phenotype Ontology database [58] was then used to generate mouse phenotype information associated with our candidate genes, which is helpful for understanding the phenotypic trait in human disease. (2) Kyoto Encyclopedia of Genes and Genomes [59] (KEGG): molecular pathway analysis was carried out using the integrated database to determine how genes are networked and enriched within the pathways. (3) Gene Ontology [60,61] (GO): terms derived from molecular function, cellular component, and biological process were annotated to decipher protein–protein interaction. Only non-redundant categories were contained by selecting the most general categories in each branch of the GO DAG structure. Significance of an enrichment result was set at *q*-value (FDR) <0.05 for all functional annotations.

## Figures and Tables

**Figure 1 jpm-12-00198-f001:**
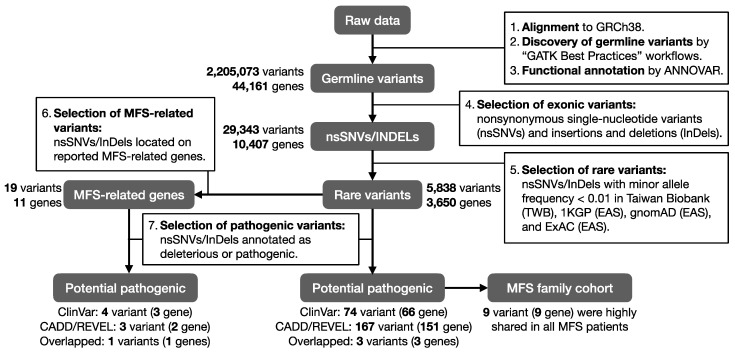
Study workflow of the MFS WES analysis.

**Figure 2 jpm-12-00198-f002:**
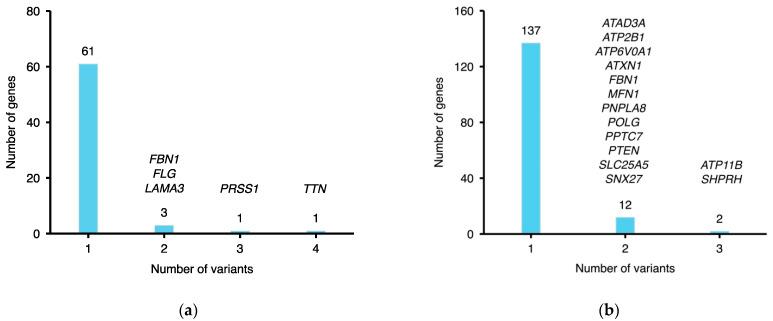
(**a**) MFS pathogenic rare variants predicted by ClinVar; (**b**) MFS pathogenic rare variants predicted by CADD and REVEL scores.

**Figure 3 jpm-12-00198-f003:**
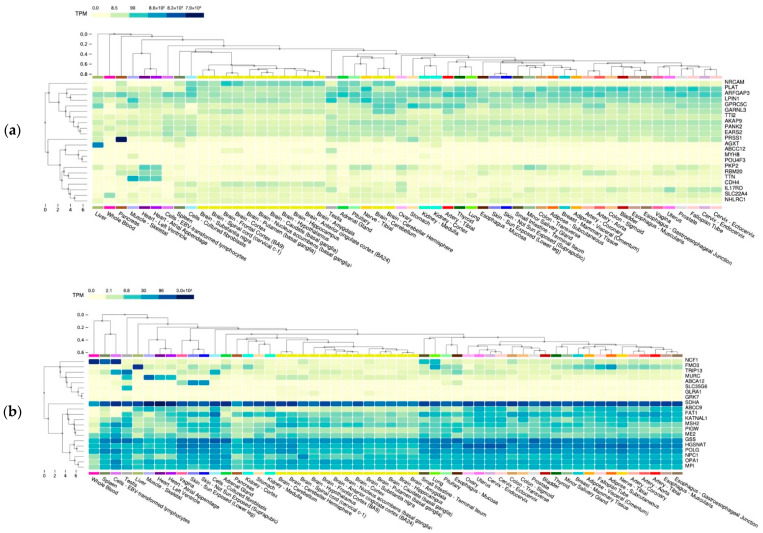
(**a**) Expression of MFS pt.2 candidate genes among all tissues.; (**b**) Expression of MFS pt.5 candidate genes among all tissues.

**Table 1 jpm-12-00198-t001:** Basal characteristic of MFS patients.

Sample	Gender	Age	Relatedness	Note	Systemic Score
pt.1	F	52	-	-	9
pt.2	F	42	-	-	5
pt.3	F	40	-	-	4
pt.4	M	43	-	-	7
pt.5	M	42	-	-	7
pt.6	M	37	-	-	4
pt.7	F	52	Family 1-Mother	-	4
pt.8	M	23	Family 1-Son-01	-	6
pt.9	M	26	Family 1-Son-02	-	4
pt.10	M	-	Family 1-Father	Healthy control	-
pt.11	F	20	Family 1-Daughter	-	7

**Table 2 jpm-12-00198-t002:** Alternative allele of the 19 rare variants on MFS-related genes in MFS patients.

Chr.	Position	Ref.	Alt.	Gene	Type	Number of Alt. Allele in MFS Patients										
						1	2	3	4	5	6	7	8	9	11	10 ^b^
1	11794020	G	A	MTHFR	nsSNV	1	-	-	-	-	-	-	-	-	-	-
2	189051324	T	C	COL5A2	nsSNV	1	-	-	-	-	-	-	-	-	-	-
3	30623240	A	C	TGFBR2	nsSNV	-	-	-	1	-	-	-	-	-	-	-
9	99105216	C	T	TGFBR1	nsSNV	-	-	-	-	-	-	-	-	-	-	1
**9**	**134758250**	**G**	**A**	**COL5A1**	**nsSNV**	**-**	**-**	**-**	**-**	**-**	**1**	**-**	**-**	**-**	**-**	**-**
11	65557857	CCAG	CCAGCAGCAG,C	LTBP3	Ins (NF)	-	1	-	-	-	1 ^a^	-	-	-	-	-
12	57167041	C	T	LRP1	nsSNV	-	-	-	-	-	1	-	-	-	-	-
15	48420690	C	T	FBN1	nsSNV	1	-	-	-	-	-	-	-	-	-	-
**15**	**48436991**	**C**	**A**	**FBN1**	**nsSNV**	**-**	**-**	**-**	**-**	**-**	**-**	**1**	**1**	**1**	**1**	**-**
15	48437824	G	T	FBN1	nsSNV	-	-	-	1	-	-	-	-	-	-	-
15	48448860	C	T	FBN1	nsSNV	-	-	1	-	-	-	-	-	-	-	-
15	48487321	GC	G	FBN1	Del (F)	-	-	-	-	-	1	-	-	-	-	-
**15**	**48503803**	**GAC**	**G**	**FBN1**	**Del (F)**	**1**	**-**	**-**	**-**	**-**	**-**	**-**	**-**	**-**	**-**	**-**
**16**	**15714999**	**T**	**C**	**MYH11**	**nsSNV**	**-**	**-**	**-**	**-**	**-**	**-**	**1**	**-**	**1**	**-**	**-**
17	63494011	G	T	ACE	nsSNV	-	-	1	-	-	-	-	-	-	-	-
17	63497343	G	T	ACE	stop gain	-	-	1	-	-	-	-	-	-	-	-
19	8123973	C	T	FBN3	nsSNV	-	-	1	-	-	-	-	-	-	-	-
19	8133099	CGTT	C	FBN3	Del (NF)	-	-	-	-	-	-	-	-	-	-	1
**19**	**8136499**	**A**	**G**	**FBN3**	**nsSNV**	**-**	**-**	**1**	**-**	**-**	**-**	**-**	**-**	**-**	**-**	**-**

^a^ For the second allele listed in Alt. ^b^ Healthy subject. Pathogenic variants are shown in bold.

**Table 3 jpm-12-00198-t003:** Functional annotations of MFS candidate genes.

Functional Annotations		No. of Reference Genes in the Category	No. of MFS Candidate Genes in the Category	*p*-Value	*q*-Value (FDR)
**Knockout Mouse Phenotype Category**					
	Abnormal cardiovascular system morphology	1794	55	1.40 × 10^−7^	9.11 × 10^−4^
	Abnormal sarcomere morphology	60	9	3.13 × 10^−7^	1.02 × 10^−3^
	Premature death	952	34	2.80 × 10^−6^	6.09 × 10^−3^
	Kyphosis	162	12	7.80 × 10^−6^	1.27 × 10^−2^
	Abnormal heart morphology	1297	40	1.17 × 10^−5^	1.49 × 10^−2^
	Abnormal blood vessel morphology	1070	35	1.38 × 10^−5^	1.49 × 10^−2^
	Abnormal stria vascularis morphology	40	6	3.16 × 10^−5^	2.92 × 10^−2^
	Ascending aorta aneurysm	5	3	3.59 × 10^−5^	2.92 × 10^−2^
**GO term Category**					
	Sensory perception of light stimulus	209	14	7.25 × 10^−7^	9.45 × 10^−4^
	ATPase activity	438	18	2.09 × 10^−5^	1.01 × 10^−2^
	Extracellular structure organization	400	17	2.33 × 10^−5^	1.01 × 10^−2^
	Contractile fiber	226	12	4.70 × 10^−5^	1.53 × 10^−2^
	Sensory system development	355	15	7.60 × 10^−5^	1.98 × 10^−2^
	Extracellular matrix	496	18	1.04 × 10^−4^	2.27 × 10^−2^
	Actinin binding	39	5	1.52 × 10^−4^	2.83 × 10^−2^
	Extracellular matrix structural constituent	158	9	2.50 × 10^−4^	3.54 × 10^−2^
	Vacuolar membrane	397	15	2.58 × 10^−4^	3.54 × 10^−2^
	Structural constituent of muscle	44	5	2.71 × 10^−4^	3.54 × 10^−2^
	Organic hydroxy compound metabolic process	500	17	3.49 × 10^−4^	3.93 × 10^−2^
	Urogenital system development	326	13	4.00 × 10^−4^	3.93 × 10^−2^
	Cell junction organization	285	12	4.11 × 10^−4^	3.93 × 10^−2^
	Multicellular organismal signaling	170	9	4.29 × 10^−4^	3.93 × 10^−2^
	Protein kinase C binding	49	5	4.52 × 10^−4^	3.93 × 10^−2^

**Table 4 jpm-12-00198-t004:** Rare variants on MFS candidate genes *POMT1*, *FBN1*, and *UNC13D* predicted by ClinVar and CADD/REVEL.

Chr.	Position	Ref.	Alt.	Gene	Type	avSNP150	Allelic Frequency				Pathogenicity			MFS ^a^
							1KGP	gnomAD	ExAC	TWB	ClinVar	CADD	REVEL	
							(EAS)	(EAS)	(EAS)					
9	131513282	G	A	POMT1	nsSNV	rs146869947	0.001	0.0019	0.0027	0.003	Conflicting interpretations of pathogenicity	26.9	0.836	7, 9, 11
15	48448860	C	T	FBN1	nsSNV	-	-	-	-	-	Pathogenic	32	0.988	3
17	75831135	C	T	UNC13D	nsSNV	rs140184929	0.006	0.0054	0.0043	0.003	Conflicting interpretations of pathogenicity	27.9	0.933	4

^a^ Patients who carry the variant.

**Table 5 jpm-12-00198-t005:** Rare variants on MFS candidate genes shared in MFS family cohort.

Chr.	Position	Ref.	Alt.	Gene	Type	Allelic Frequency	Alt. Allele of				
							a MFS Family				
						TWB	7	8	9	11	10 ^a^
2	178794954	C	T	TTN	nsSNV	0.007	1	-	1	1	-
3	52199279	G	A	ALAS1	nsSNV	0.0005	1	1	1	1	-
3	107716694	A	G	BBX	nsSNV	-	1	1	-	1	-
9	131513282	G	A	POMT1	nsSNV	0.003	1	-	1	1	-
12	109264325	C	T	ACACB	nsSNV	-	1	1	-	1	-
15	40856938	A	G	SPINT1	nsSNV	-	1	-	1	1	-
15	48436991	C	A	FBN1 ^#^	nsSNV	-	1	1	1	1	-
20	45895081	G	A	CTSA	nsSNV	0.004	1	1	1	-	-
20	63350423	C	T	CHRNA4	nsSNV	-	1	-	1	1	-

^a^ Healthy subject. ^#^ Previously reported MFS-related genes.

## Data Availability

Raw data were generated at Taipei Medical University. Derived data supporting the findings of this study are available from the corresponding author on request.

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
