# Peer review of "Application of Whole Exome Sequencing and Functional Annotations to Identify Genetic Variants Associated with Marfan Syndrome"

_jpm, 2022, doi:10.3390/jpm12020198_

Round 1

Reviewer 1 Report

The authors aimed to identify genetic factors associated with Marfan Syndrome in a Tiawanese patient sample using whole exome sequencing. However, based on the information presented, it does not appear that this is what they did.  There are several items in the methodology and conclusions in this analysis that require significant revision before it can be considered for potential publication.

The largest limitation is that this is a sample of 10 patients and 1 control. Of these 10 patients, 5 are related, and the control is also related to the other 5 family members.  This is not an appropriate design for meaningful conclusions for gene-disease relationship claims.  Particularly, the control data is essentially irrelevant due to his biological relationship with several of the affected individuals.  The authors acknowledge this as a limitation, but the scope of not appropriately handled in their work.

Further, there is essentially no discussion of the phenotypic adjudication of the individuals.t is unclear if they even meet marfan syndrome criteria.  Moreover, it is difficult to discern is even related specifically to the strictly defined marfan syndrome phenotype or if a pursuit of any genes related to connective tissue diseases, aortopathies (syndromic or non-syndromic), including   This is further emphasized in their "Marfan related gene list" with an absence of methods or rationale for the gene listthe defined as "marfan related genes." Although not made transparent, my take  broader investigation than marfan syndrome but without the phenotypic inclusion criteria for the study and methods for how the related gene list was curated, it is impossible to tell.

Another major concern is the unfounded statements of causality of other cardiovascular genes in association with this highly specific marfan phenotype. The authors provide several overstatements suggesting causality based on cardiac protein expression alone. Specifically, TTN variants are well known to be over represented in a healthy population. They provide no additional data or rational on the relationship with TTN and marfan, nor do they do this for the other candidate genes aside from the fact that the variants were rare and may in some way contribute to cardiac tissue disease. This also completely ignores the relationship of these genes with the many other systemic features that are classic to marfan syndrome.

I have several other more granular comments about the manuscript, but until the larger framing and approach of this paper is clarified, I do not think these additional items are appropriate to provide at this stage.

Reviewer 2 Report

Lin et al. present a study in which they collected 10 cases of Marfan syndrome and one healthy control. Among all subjects, one family is included, consisting of four members. The aim of this study was to genetically characterize cases with Marfan syndrome and to detect potential novel disease-associated genes using whole exome sequencing. In order to annotate candidate genes, a functional annotation was used, consisting of knockout mouse phenotype category and gene ontology category. In two of the ten cases, the mutation in FBN1 was detected, which is expected since FBN1 pathogenic variants are known to be associated with Marfan. Functional annotation further revealed novel candidate genes (TTN and POMT1) which could play a functional role in disease development. Analysis of family members revealed potentially novel pathogenic variants in FBN1 gene.   

The manuscript is well-written and clear with somehow limited novelty. Additionally there are some issues, which need to be addressed:

  1. Line 30: “we verified the pathogenicity of FBN1 to MFS”. I believe that the role of FBN1 gene in MFS is very well established in different populations so there is no need for “verifications”.

  1. Line 30: “and reported TTN and POMT1 to be critical novel disease-associated genes”. I believe that based on the design and the results of this study, this statement is overrated.

  1. Table 2. I believe that there is an error in all allele frequencies (AF). The decimal point is wrongly positioned. I don’t think that the patients have only 4.5% (0.045) alternative allele frequency, but I guess the correct value is 45% (0.45). If there is indeed such a low AF, then it should be explained.

  1. Figure 2a. Orientation of the graphic is strange.

  1. Table 2, Table 4, Table 5. Tables should have both alleles (normal and alternative allele).

Limitations of the study: as authors already discussed, a larger sample size is needed to reliably conclude about genotype-phenotype correlations. In order to reliably detect candidate genes, more families would be needed in order to perform segregation analysis and confirm associations.

Round 2

Reviewer 1 Report

The responses from the authors is appreciated, however, it does not appear that many changes to the manuscript addressing the key issues raised were made.  Below are my major and minor suggestions for improvement to consider.

Major:

  • While the authors provide a new sentence stating that a the ghent nosology were used by physicians to make the Marfan diagnosis, the breakdown and transparency of these data are unclear. Table 1 should include what system score features were met by each subject, aortic root size, and z score, etc to show exactly how each individual met the Marfan diagnosis.  
  • Claiming that TTN, RBM20, and other cardiomyopathy genes have a role in Marfan syndrome is still over suggested. If these claims are going to be made, minimally providing clinical data to demonstrate that there was no suggestion of cardiomyopathy in these individuals would be at least a bit helpful to show that these variants are not causing a cardiomyopathy or other established cardiovascular phenotype that has been associated with these genes. Further, the authors may wish to consider a multi-variant hypothesis for the non-FBN1 cases. 
  • Line 170: The authors state that the “two variants carried by ALAS1 and FBN1 were shared in all four MFS patients, suggesting these genes are crucial in the pathogenesis of MFS.” However, without any data supporting the individual biological relevance of the ALAS1 variant (in cases without other highly plausible disease causing variants, such as in FBN1 in this family), this really isn’t suggested at all. In this section of the results, I would suggest simply stating that variants in both of these genes were seen in all four patients. The possible/lack of relationship of ALAS1 with a marfan phenotype is addressed in the first paragraph of the discussion, which is sufficient.
  • Please consider consulting the ClinGen gene curation work for aortopathies and marfan syndrome to ensure that statements regarding gene-disease relationships throughout the manuscript are as appropriately stated. 

      • PMID: 30071989
    https://search.clinicalgenome.org/kb/conditions/MONDO:0007947

    Minor:

    -Line 50: "Diagnositic" should be "Diagnosis"

    -Use the term "Variant" instead of "mutation" throughout the manuscript. This is more contemporary nomenclature.

    -I am familiar with and understand the elated family members for controls, the issue here in is that there is just one individual participating in this way.  Accordingly, in lines 102-103, stating that the FBN1variant absence in the healthy control parent "confirms" this finding still feels sounds inflated.  Perhaps "suggests" as a less certain alternative word choice.

    -Table 4: Title is not descriptive for what is being shown. Titleing “Marfan Syndrome Pathogenic Rare Variants..” and then going on to list that there are “conflicting interpretations of pathogenicity” is confusing and misleading. This same recommendation iapplies to Table 5 title.

    -Table 4: Please clarify if the allele frequencies in table 4 are percentages?

    -Line 229: italicize TTN.

    -Line 230: change “sarcomeric” to sarcomere.

    Author Response

    This manuscript is a resubmission of an earlier submission. The following is a list of the peer review reports and author responses from that submission.